

# Liver slice culture as a model for lipid metabolism in fish

Thomas N. Harvey[1], Simen R. Sandve[1], Yang Jin[1], Jon Olav Vik[1] and Jacob S. Torgersen[2]

[1] Centre for Integrative Genetics, Department of Animal and Aquacultural Sciences, Faculty of Biosciences, Norwegian University of Life Sciences, Ås, Norway
[2] AquaGen AS, Trondheim, Norway

## ABSTRACT

Hepatic lipid metabolism is traditionally investigated in vitro using hepatocyte monocultures lacking the complex three-dimensional structure and interacting cell types essential liver function. Precision cut liver slice (PCLS) culture represents an alternative in vitro system, which benefits from retention of tissue architecture. Here, we present the first comprehensive evaluation of the PCLS method in fish (Atlantic salmon, Salmo salar L.) and validate it in the context of lipid metabolism using feeding trials, extensive transcriptomic data, and fatty acid measurements. We observe an initial period of post-slicing global transcriptome adjustment, which plateaued after 3 days in major metabolic pathways and stabilized through 9 days. PCLS fed alpha-linolenic acid (ALA) and insulin responded in a liver-like manner, increasing lipid biosynthesis gene expression. We identify interactions between insulin and ALA, where two PUFA biosynthesis genes that were induced by insulin or ALA alone, were highly down-regulated when insulin and ALA were combined. We also find that transcriptomic profiles of liver slices are exceedingly more similar to whole liver than hepatocyte monocultures, both for lipid metabolism and liver marker genes. PCLS culture opens new avenues for high throughput experimentation on the effect of "novel feed composition" and represent a promising new strategy for studying genotype-specific molecular features of metabolism.

## INTRODUCTION

Liver is the metabolic transformation hub. It is responsible for receiving nutrients absorbed in the gut through the portal vein, processing these nutrients for storage or energy production, and subsequent transportation of metabolic products throughout the body. Essential to proper function, the complex three-dimensional structure of liver consists of intrahepatic microcirculatory units (lobules) of tightly associated cells that communicate through paracrine and autocrine effects (*Wake & Sato, 2015*). The liver is also the main organ for detoxification, so in vitro methods are commonly applied for toxicological studies to reduce use of in vivo experiments. Hepatocyte cultures were established in the 1970s (*Ekins, 1996*), and rapidly became the preferred model system for toxicology. Liver slice culture was first introduced in 1923 (*Warburg, 1923*), but seldom used due to a lack of reproducibility since slices needed to be cut by hand. The development of

Corresponding authors
Jon Olav Vik, jonovik@gmail.com
Jacob S. Torgersen,
jactor@aquagen.no

automated tissue slicers in the 1980s (*Krumdieck, Dos Santos & Ho, 1980*) solved this problem, so liver slices became a viable option. One of the main advantages of liver slices is the retention of normal cell composition and 3D structure. In addition, the preparation is fast, reproducible, without enzymatic cell dissociation, and no need for coating the growth surface. Together with established protocols, this has heralded the return of liver slices for in vitro studies.

Precision cut liver slice (PCLS) cultures have been applied in a number of toxicology studies and most recently also immunology (*Wu et al., 2018*), however, the use of PCLS to study central liver metabolism is sparse, with few PCLS studies investigating aspects of lipid metabolism, all of which are in mammals (*Neyrinck, Gomez & Delzenne, 2004*; *Szalowska et al., 2014*; *Janssen et al., 2015*; *Fortin et al., 2017*). We provide a critical evaluation of PCLS as a metabolic model system in fish by characterizing whole transcriptome changes in the context of lipid metabolism. We chose Atlantic salmon for its economic importance; and because development of feeding and breeding strategies that optimize omega-3 production require a better understanding lipid metabolism. Additionally, the effect of altered feed fatty acid profile on liver gene expression is well documented (*Tocher et al., 2001*; *Leaver et al., 2008*; *Gillard et al., 2018*) making this an ideal system for assessing the effects of altering media fatty acid composition and comparing to expected in vivo gene expression.

Here, we integrate transcriptomics data with domain knowledge to describe a method for using PCLS as a model system to study lipid metabolism. We aim to (1) characterize transcriptome wide changes in liver slice culture over time, (2) demonstrate the utility of using liver slice culture to study lipid metabolism, and (3) compare gene expression patterns between liver slice culture, 2D hepatocyte culture, and whole liver in vivo.

## MATERIALS AND METHODS

### Liver slice culture

Atlantic salmon used in this study were sacrificed according to the Norwegian Animal Research Authority; regulations for use of experimental animals (FOR-2015-06-18-761). The liver was removed immediately after euthanization and placed in ice cold Hank's balanced salt solution (HBSS; Thermofisher, Waltham, MA, USA). Livers were cut into approximately $4 \times 4 \times 8$ mm strips before being superglued to a plastic piston and encased in ultra-low melt agarose (Merck KGaA, Darmstadt, Germany). Liver strips were sliced to a thickness of 300 µm using a compresstome VF-300 (Precisionary Instruments, Greenville, NC, USA) and collected in ice cold HBSS before being transferred to 15 °C Leibovitz 15 medium pH 7.4 (L15; Thermofisher) containing 5% fetal bovine serum (FBS; Merck) and 1% penicillin—streptomycin (Thermofisher) which will now be referred to as base media. Liver slices were incubated in sterile 6 (four mL media per well) or 12 (two mL media per well) well cell culture plates with netwell inserts (Corning Netwell 500 µm memebrane, Merck) for up to 9 days at 15 °C under ambient air. For each of the following experiments, liver slices were prepared from a single fish each time to eliminate biological variation.

## Time course experiments

We performed two-time course experiments, the first to test the effect of culturing time on the liver slices, and the second as a follow up to test the effect of media change frequency and inclusion of insulin over time. In both experiments, liver slices were generated immediately after euthanization and viability measurements were taken every day in the first experiment and on days 3 and 6 in the second experiment. All samples were stored in RNAlater at $-20\,°C$. In the first experiment we generated slices from a saltwater life-stage Atlantic salmon (~200 g) reared on a marine oil-based diet high in DHA and EPA. Immediately after euthanization, liver slices were generated as described above. Media was changed after sampling on days 3 and 6 using base media. On day 3 base media was supplemented with 700 µM randomly methylated beta-cyclodextrin (BCD) and 0.7% ethanol. Samples were taken in triplicate before slicing (whole liver) and 1, 3, 4, 5, 6, 7, 8, and 9 days after slicing. In the second time course experiment we used Atlantic salmon in the freshwater life-stage reared on a marine oil diet high in EPA and DHA. Human insulin (Merck) was included in the media at 20 nM and media was refreshed either every day or every 3rd day with fresh base media containing 20 nM insulin. Samples were taken in triplicate before slicing (whole liver) and days 3, 4, 5, 6, 7, 8, and 9.

## Fatty acid and insulin gradient experiments

We performed two concentration gradient experiments, the first was used for transcriptomic analysis, the second for fatty acid analysis. In the first experiment liver slices were prepared from two freshwater stage Atlantic salmon (~50 g), one for use in the fatty acid gradient experiment and one for use in the insulin gradient experiment. We used randomly methylated BCD as our fatty acid delivery system since it has been demonstrated to efficiently deliver fatty acids across membranes in other in vitro systems (*Brunaldi, Huang & Hamilton, 2010*). Alpha-linolenic acid (ALA) was stored at 10 mM in ethanol then mixed 1:1 with 100 mM BCD in water for a final molar ratio of 1:10 fatty acid to BCD. From this stock ALA was added to the media at a concentration of 0 (700 µM BCD only), 20, 40, 70, and 100 µM, aliquoted into a new six well culture plate, and placed at $15\,°C$ to equilibrate for at least 30 min. For all ALA treated samples, liver slices were transferred to ALA supplemented base media after a 3-day recovery period. For the insulin containing samples, human insulin (Merck) was diluted in base media to a final concentration of 10 or 100 nM and incubated with liver slices from the beginning of the experiment. All liver slices were sampled in triplicate on day 5 and stored in RNAlater at $-20\,°C$. In the second concentration gradient experiment, liver slices were prepared from freshwater stage fish (~500 g) and supplemented with 0 (700 µM BCD only), 20, 40, 70, 100, and 140 µM ALA on day 3 as described except this time ethanol was evaporated under a stream of nitrogen before mixing with BCD. Samples were taken in triplicate on day 4, washed in ice cold HBSS, flash frozen in an ethanol dry ice slurry, and stored at $-80\,°C$.

## 2D hepatocyte culture experiment

Primary cells were isolated from salmon liver as described (*Bell et al., 1997*), with some modifications. After euthanization, the liver was removed and rinsed in ice cold $Mg^{2+}/Ca^{2+}$

free HBSS, before ~100 mL of the same buffer was directly injected with a 50 mL syringe and 27 G needle, at various places to wash out blood cells. Then, 30 mL of HBSS with 150 U/mL Collagenase type 1 (Merck) was injected, before the tissue was finely chopped. The tissue suspension was incubated for 1 h at 10–12 °C with agitation. Dissociated cells were collected by cell straining (70 µm) and centrifugation for 10 min at 100 g. After three washes in HBSS, the pellet was resuspended in base media supplemented with 10 µM human insulin (Merck) and grown at 200 k/cm$^2$ density at 15 °C on polyethyleneimine coated wells (*Vancha et al., 2004*). Cells were supplemented with ALA on day 5 as previously described and collected in triplicate using a cell scraper on days 5 (before ALA), 6, and 8 by flash freezing and storing at −80 °C.

## Viability measurement

Slice viability was assessed by staining with Hoechst and propidium iodide to identify live and dead cells. Slices were transferred to L15 medium containing 10 µg/mL Hoechst and 10 µg/mL propidium iodide for 5 min at 15 °C. Slices were then transferred to fresh L15 medium and placed on ice until being imaged with a scanning laser confocal microscope (Leica, Wetzlar, Germany). Live/dead ratios were determined using Icy (http://www.bioimageanalysis.org/). We compared the proportions of live and dead cells in two randomly selected locations per slice to determine overall slice viability.

## Microscopy

We made cross sections of liver slices at three different time points during culturing (one slice per time point) and observed morphological changes using light microscopy. All samples for microscopy were fixed using 4% formalin in phosphate buffered saline (PBS) for 1 h then transferred to 70% ethanol stepwise (PBS-25–50–70%) for 5 min at each step and stored at −20 °C until microscopic analysis was performed. Prior to paraffin embedding liver slices were transferred to 96% ethanol stepwise (70–85–96–96%) for 5 min at each step then washed twice with histoclear (National Diagnostics, Atlanta, Georgia, USA) for 5 min each. Next, liver slices were embedded in paraffin (Merck) by incubating in paraffin at 61 °C three times for 10 min each. Paraffin was allowed to solidify at room temperature. Liver slice cross-sections were prepared using a rotary microtome (Leica, Wetzlar, Germany) at a thickness of seven µm, placed on the surface of a 43 °C water bath, and floated onto a clean microscopy slide. Sections were deparaffinized by washing twice with histoclear for 5 min each and rehydrated by transferring to 70% ethanol stepwise (histoclear-96–85–70–70%) for 5 min each followed by a brief wash in distilled water. Sections were stained with a 1% hematoxylin solution (Mayer's) for 8 min, rinsed in running tap water for 10 min followed by 96% ethanol and counterstained with a 0.25% eosin-phloxine B solution for 30 s. Stained sections were washed twice with histoclear for 5 min each and mounted with DPX (Merck). Micrographs of cross-sections were taken at 20× magnification on a light microscope (Leica, Wetzlar, Germany).
## RNA sequencing

Slices were stored in RNAlater (Merck) at −20 °C until RNA extraction using the RNeasy universal kit (QIAGEN, Hilden, Germany). RNA concentration was determined on a Nanodrop 8000 and quality was determined on an Agilent 2100 bioanalyzer using Agilent RNA 6000 nano chips. All RNA samples had a RNA integrity number greater than 7. mRNA libraries were prepared using the Trueseq library preparation kit (Agilent, Santa Clara, CA, USA). Concentration and mean length were determined by running cDNA libraries on a bioanalyzer 2100 using DNA 1000 chips (Agilent). RNA libraries were sequenced on an Illumina HiSeq 2500 with 100-bp single end reads.

## RNAseq analysis

All RNA sequencing and demultiplexing was done at the Norwegian sequencing center (Oslo, Norway). Fastq files were trimmed and mapped the salmon genome (ICSASG_v2) using STAR (v2.5.2a) (*Dobin et al., 2013*). Mapped reads for each gene were counted with HTSeq-count (v0.6.1p1) (*Anders, Pyl & Huber, 2015*). Differential expression analysis was performed in R (v3.2.5) using the edgeR package (*Robinson, McCarthy & Smyth, 2010*). All counts were normalized to library size using TMM normalization within edgeR. For the time course and gradient experiments an analysis of variance analysis of variance (ANOVA)-like differential expression test was used to find difference between any of the conditions (see edgeR manual). This yielded log2 fold change to the reference level (day 0 or ALA 0) and false discovery rate (FDR) for each gene. For the time course experiments we considered genes with a FDR of <0.01 and log2 fold change (log2FC) > 1 as differentially expressed while for the gradient experiments genes with a FDR of <0.01 were considered differentially expressed. Gene expression clusters were generated by applying wardD2 hierarchical clustering to gene-scaled mean counts per million (CPM). Kyoto encyclopedia of genes and genomes (KEGG) enrichment was performed on each gene cluster using edgeR. Pathways with a $p$-value < 0.001 were considered significantly enriched. To compare gene expression between whole liver, liver slice, and hepatocyte culture, we pooled data from each source to give an overall expression phenotype. Data on whole liver was obtained from a previously published feeding trial (*Gillard et al., 2018*) and whole liver samples taken before generating liver slices. Data on liver slice and hepatocyte culture was obtained from the previously described experiments.

## Lipid analysis

Fatty acid methyl esters (FAME) were prepared from liver slices according to established protocols (*O'Fallon et al., 2007*) with half volumes to account for the small size of liver slices. We used 13:0, 19:0, and 23:0 as an internal standards in all samples and FAMEs were separated by gas chromatography on a Trace GC Ultra (Thermo Fisher, Waltham, MA, USA) using a flame ionization detector. Relative fatty acid abundance was calculated from the resulting chromatograms.

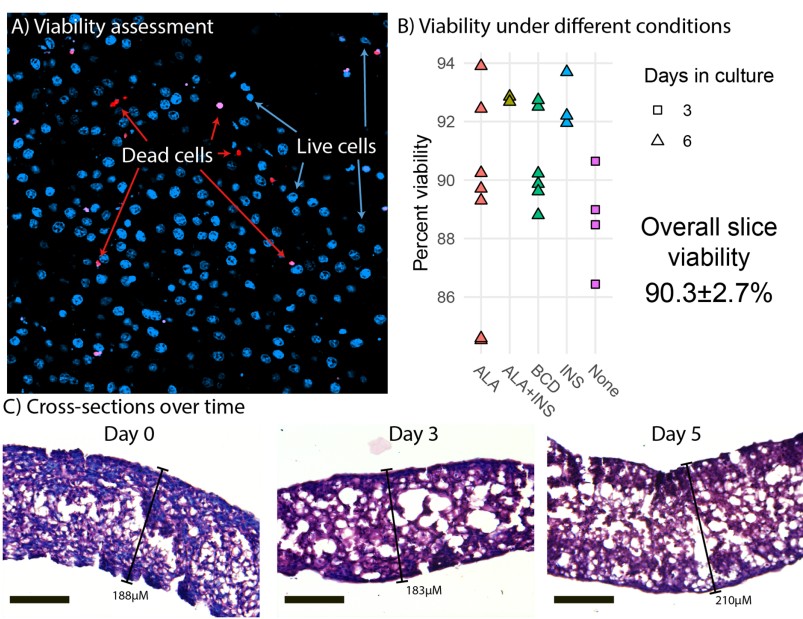

**Figure 1 Viability and morphology of liver slices.** (A) Confocal scanning laser microscope (CLSM) image of a liver slice. Cells are stained with Hoechst (blue) and dead cells with propidium iodide (red). (B) Cell viability when incubated in media (none) and media supplemented with alpha-linolenic acid (ALA), insulin (INS), empty methyl-β-cyclodextrin (BCD), or a combination as measured by CLSM live/dead counts. Points represent viability measurements from a single liver slice. (C) Cross sections of paraffin embedded liver slice sampled at day 0 (immediately after slicing), day 3, and day 5. The sections were stained with hematoxylin and eosin and photographed using light microscopy at 20× magnification. Scale bars are 100 μM in length.

## Statistical analysis

All statistical analysis was performed in R (v3.2.5). Correlation analysis between whole liver and liver slice samples was calculated using the mean CPM of each gene across the three-time course experiments (whole liver) and triplicate samples within each experiment (liver slice) for each day followed by Spearman's rank correlation test. Comparison of gene expression (CPM) between groups in the ALA and insulin gradient experiments was calculated using a one-way ANOVA test followed by a Tukey–HSD test. Differences with a $p$-value < 0.05 were considered significant.

## RESULTS

### Viability and morphology

Liver slices had a mean viability of 90.3% ± 2.7% (Fig. 1B). We did not observe any viability effects of ALA, insulin nor methyl-BCD used as a lipid carrier in the experiments. More generally, we find that viability at the end of an experiment is similar to the viability at the beginning of an experiment (Fig. 1B). This implies that preparation of the slices is most critical to viability, as opposed to culture time. Morphological analysis of liver slice cross-sections did not reveal any large change in the thickness of slices over a 5-day period. Slices did appear thinner than they were cut (300 μM), but this is likely due to dehydration causing the liver slices to shrink during the paraffin embedding process

(Fig. 1C). Slices examined were approximately 188, 183, and 210 μm in thickness on days 0, 3, and 5, respectively.

## Time course experiments

In order to study how the liver slices change in culture over time, we sequenced RNA from three experiments lasting for 9 days. In time course one, media was changed every 3 days and samples were taken before slicing (day 0) and 1, 3, 4, 5, 6, 7, 8, and 9 days after slicing. On day 3, slices were fed a control diet consisting of BCD only. We use BCD to deliver the fatty acids to the cells, so in this case BCD only was used as a control for fatty acid supplementation conditions. The second- and third-time course experiments differed from the first in terms of media change frequency (daily or every 3rd day) and inclusion of insulin in the media (20 nM).

To characterize the behavior of liver slices over time under control conditions, we performed ANOVA-like differential expression analysis testing for differentially expressed genes (DEG) between any of the time points in time course one. This yielded 16,267 DEG with a FDR < 0.01 and a log2FC > 1 (Fig. 2A). We used hierarchical clustering to group genes with a similar expression trend into eight gene clusters (Fig. 2B), then searched for enriched pathways from the KEGG in each of these clusters ($p < 0.0001$) to characterize the overall trend of various physiological and metabolic processes (Fig. 2C). Interestingly, almost all pathways related to protein, lipid, carbohydrate, and vitamin metabolism belong to clusters two and three, which decreased between day 0 (before slicing) and day 3, followed by an overall stabilization in expression through day 9. Pathways related to signal transduction were mostly enriched in clusters seven and eight, which increased expression greatly between day 0 and day 1 (before and 24 h after slicing), then decreased to original levels by day 9. Pathways related to cell growth and death were mostly enriched in clusters four, five, and six, which in general increased during 9 days of liver slice culture.

Since cell culture aims to mimic the conditions and behavior of tissue in vivo, we compared gene expression patterns between whole liver and liver slices for all three experiments. To assess the similarity in expression patterns over time we calculated Spearman co-expression correlations between mean whole liver gene expression and gene expression from each day in three-time course experiments for all genes and genes within seven relevant lipid metabolism pathways (Fig. 3). For time course one, correlation between liver slices and whole liver decreased gradually over time from 0.90 on day 1 to 0.83 on day 4, then stabilized around 0.8 through day 9. A similar effect was observed in time course two and time course three with co-expression correlation to whole liver stabilizing around 0.82 through day 8 then decreasing to 0.78 and 0.79, respectively, on day 9 (Fig. 3). The greatest difference between whole liver and liver slices was in the pathway "*Steroid biosynthesis*" with co-expression correlations hovering between 0.48 and 0.28 during days 3–9 for all three experiments, the result of upregulation of nearly all steroid biosynthesis genes in liver slices. Co-expression correlation was slightly more stable over time when media was refreshed daily, especially "*Steroid biosynthesis*"; however, overall expression similarity to whole liver was high for nearly all pathways and time points examined.
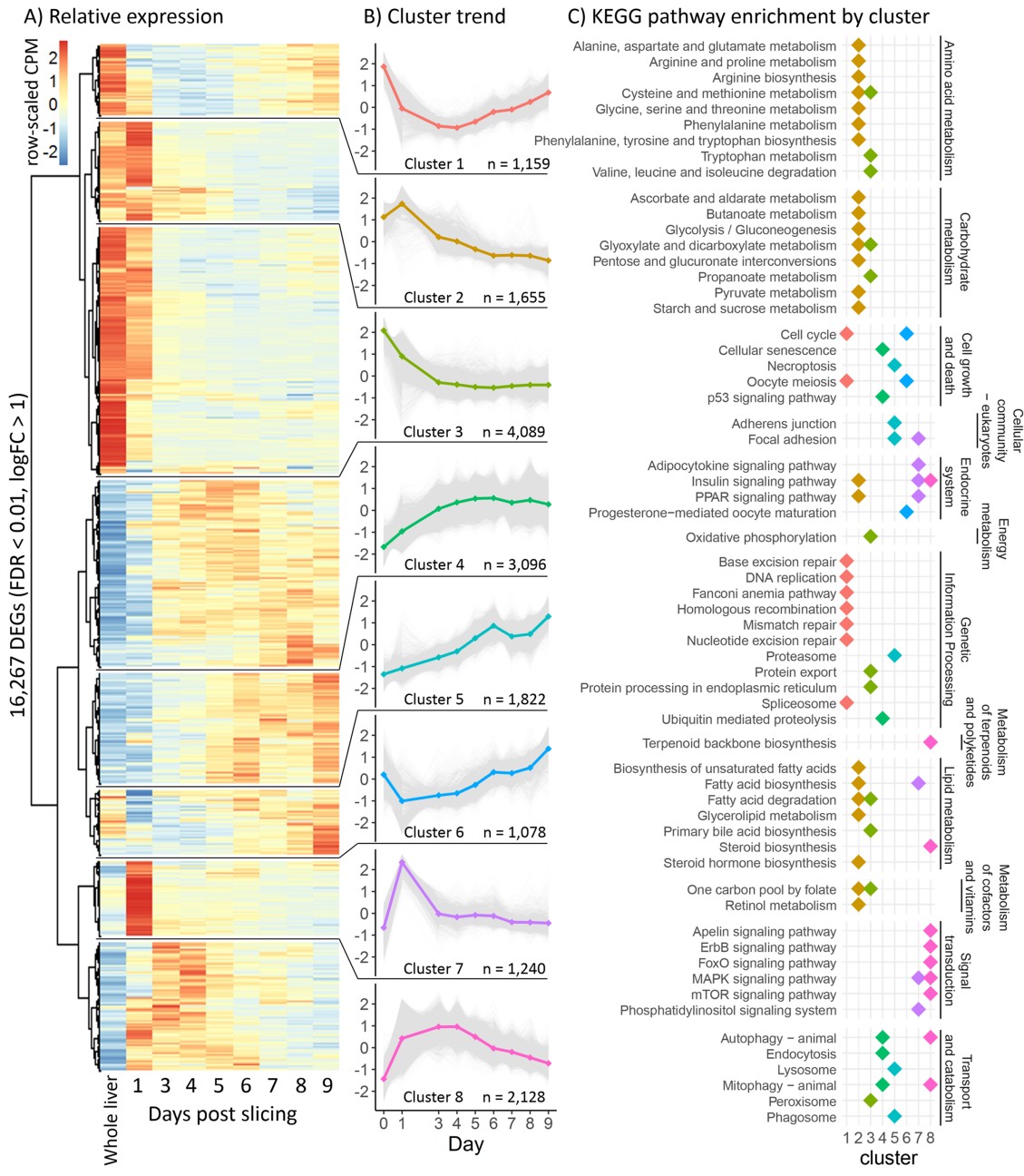

**Figure 2 Global gene expression patterns over time.** (A) Heatmap showing changes in the liver slice transcriptome over time. The heatmap includes 16,267 genes significantly differentially expressed (FDR < 0.01, log2FC > 1) over the course of 9 days. Each time point was measured in triplicate. Transcript abundance is expressed in counts per million and were individually scaled across days to highlight changes in gene expression. (B) Genes behaving similarly over time were clustered using Ward's method and broken into eight groups. Trend lines are based on mean scaled values in each cluster. (C) KEGG pathway enrichment analysis was run on each cluster to determine how the liver slices are changing over time. Each point represents a significantly enriched pathway ($p < 0.001$).

## Fatty acid and insulin gradient experiments

In order to evaluate fatty acid uptake and transcriptomic response in liver slices, we added ALA to the media in increasing concentrations from 20 μM up to 100 μM. We expect this to trigger upregulation of lipid metabolism-related gene expression as observed in liver

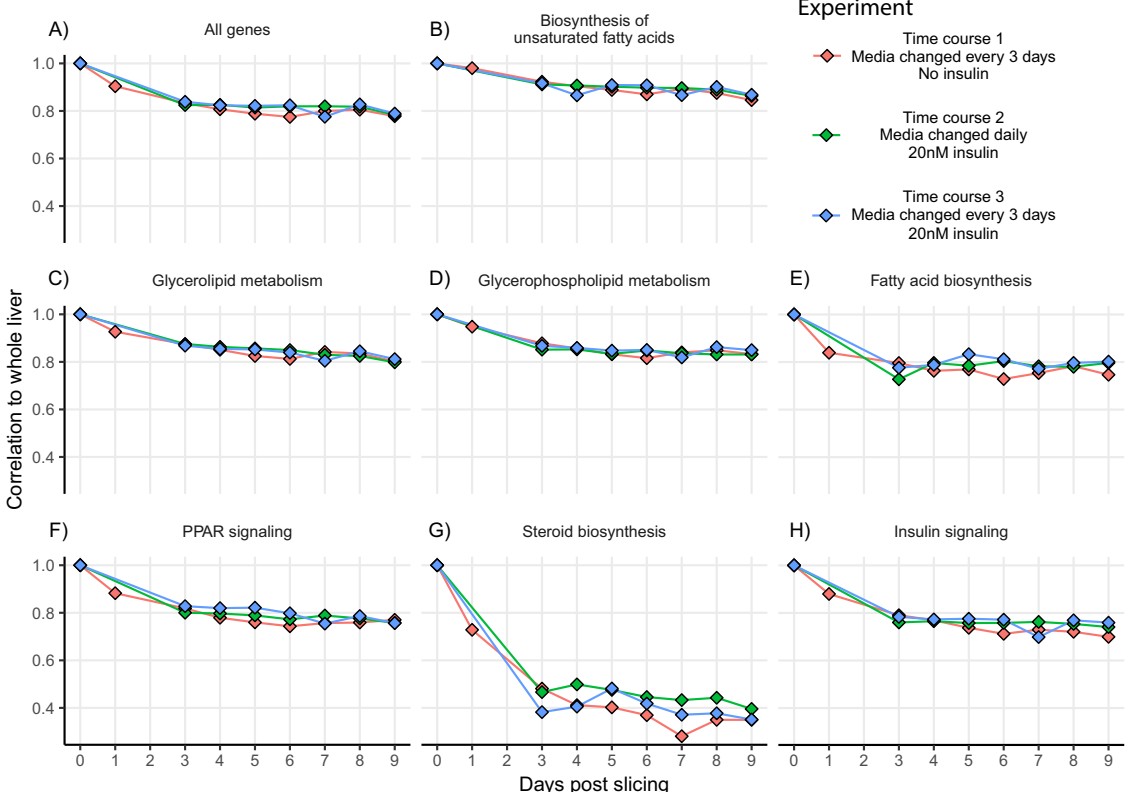

**Figure 3 Co-expression correlation of liver slices over time for select lipid metabolism pathways.** Co-expression correlations (Spearman) between mean whole liver expression and gene expression from different days in time course one (red), time course two (green), and time course three (blue). Experiments use data from liver slices in triplicate each day. Experiments were divided by insulin and media change regime. Time course two and three contained insulin, time course one and three had media changed every 3 days, and time course two had media changed daily. Correlations were calculated for all expressed genes (A) and genes from seven lipid metabolism pathways, including (B) Biosynthesis of unsaturated fatty acids, (C) Glycerolipid metabolism, (D) Glycerophospholipid metabolism, (E) Fatty acid biosynthesis, (F) PPAR signalling, (G) Steroid biosynthesis and (H) Insulin signalling.

of fish fed vegetable oil diets high in ALA (*Gillard et al., 2018*). ANOVA-like differential expression analysis testing for differences between any of the ALA concentrations yielded 8,282 DEGs (FDR < 0.01, Fig. S1A). We then broke these DEGs into four expression clusters as previously explained (Fig. S1B). KEGG enrichment analysis on these four clusters yielded 37 total pathways significantly enriched ($p < 0.001$) in one or more cluster (Fig. S1C). We found that all enriched pathways relating to lipid metabolism belonged to the same cluster which increased with increasing ALA concentration, especially between 40 and 70 µM. Specifically, the pathways "*biosynthesis of unsaturated fatty acids*," "*fatty acid degradation*," "*glycerolipid metabolism*," "*steroid biosynthesis*," and "*PPAR signaling pathway*" were all enriched in this cluster (Figs. S1B and S1C).

In order to better characterize the effect of ALA supplementation on PUFA biosynthesis, we analyzed individual gene expression of key genes in the PUFA biosynthesis pathway (Fig. 4). The five key genes involved in PUFA biosynthesis that are differentially expressed at some point in the ALA concentration gradient include delta-5 desaturase (*Δ5fad*), delta-6 desaturase a (*Δ6fada*), fatty acid elongase 2 (*elovl2*), fatty acid elongase 5a (*elovl5a*), and fatty acid elongase 5b (*elovl5b*). All five genes displayed an overall positive

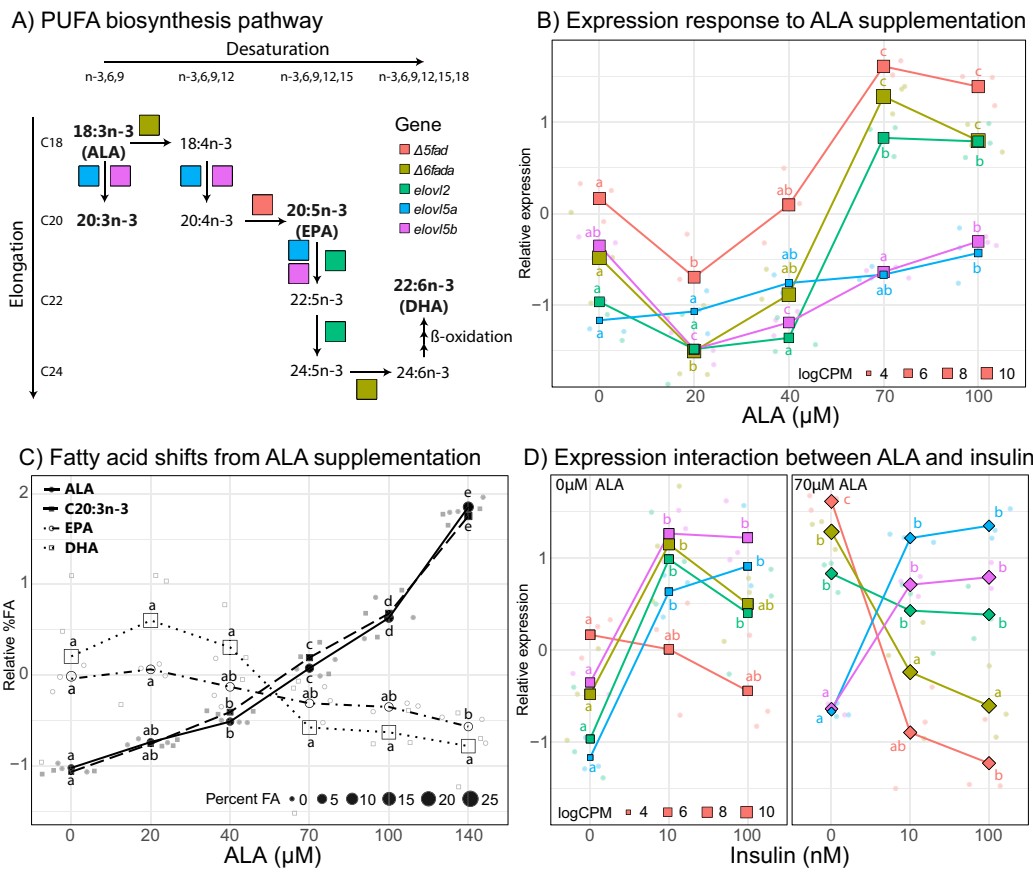

**Figure 4** **Effect of ALA and insulin on PUFA biosynthesis in liver slices.** (A) Schematic diagram of the polyunsaturated fatty acid biosynthesis (PUFA) biosynthesis pathway. (B) Gene-scaled log counts per million (CPM) of PUFA biosynthesis genes with increasing alpha-linolenic acid (ALA) concentration. (C) Gene-scaled logCPM of PUFA biosynthesis genes with increasing insulin concentration with and without ALA supplementation. (D) Relative abundance of ALA, 20:3n3, eicosapentaenoic acid, and docosahexaenoic acid with increasing ALA concentration. For all plots, large square, diamond, or circle points show mean scaled values of triplicate slices (logCPM or percent fatty acid) while small points show scaled values of individual replicates. Point size corresponds to unscaled values (logCPM or percent fatty acid) of the mean. Letters indicate significant ($q < 0.05$) differences between groups (ALA or insulin concentration) for corresponding genes or fatty acids.

correlation with ALA concentration (Fig. 4B) with *Δ5fad*, *Δ6fada*, and *elovl2* responding strongly to ALA between 40 and 70 μM and both *elovl5* genes less influenced, slightly increasing with increasing ALA concentration. Counterintuitively, at low ALA concentration (20 μM) all genes except *elovl2* and *elovl5a* significantly decreased ($q < 0.05$) in expression compared to control samples (no fatty acid). Between 70 and 100 μM ALA, expression of *Δ5fad*, *Δ6fada*, and *elovl2* did not significantly change (Fig. 4B).

To assess the impact of ALA supplementation (up to 140 μM) on the fatty acid profile of liver slices we conducted a second ALA concentration gradient experiment. As expected, percent ALA increased with increasing media ALA concentration from 0.87% with no ALA supplementation to 6.5% with 140 μM ALA supplementation (Fig. 4C). In addition, the elongation product of ALA, 20:3*n*-3, increased with increasing media ALA concentration from 0.35% with no ALA supplementation to 2.7% with 140 μM ALA

supplementation (Fig. 4C). EPA and DHA levels do not significantly ($q < 0.05$) change at any point in the ALA gradient (Fig. 4C). There was a large difference in proportions of 18:0, EPA, and DHA between fresh liver and liver slices after 4 days of incubation (Tables S1 and S2). 18:0 doubled, increasing from 5.8% in fresh liver to 12.3% in liver slices. Both EPA and DHA decreased in liver slices, from 6.4–3.5% to 29.6–25.1%, respectively (Table S1).

The effect of insulin supplementation on liver slices was assessed by incubating slices with two different concentrations of insulin, 10 and 100 nM. To test for an interaction between insulin and fatty acid supplementation, we also tested these insulin levels with and without supplementation of 70 μM ALA. Differential expression analysis testing for changes in expression between any of the conditions (without ALA supplementation) yielded 11,898 DEGs (FDR < 0.01, Fig. S2A). Approximately half of these genes were upregulated (5,889 DEGs) and half were downregulated (6,012 DEGs) regardless of insulin concentration (Fig. S2B). Only 13 genes were differentially expressed between the two insulin concentrations. KEGG pathway enrichment on these gene clusters revealed that most metabolism related pathways were upregulated with the addition of insulin. Specifically relating to lipid metabolism, "*biosynthesis of unsaturated fatty acids*," "*fatty acid biosynthesis*," and "*PPAR signaling pathway*" were significantly enriched in the upregulated gene set (Fig. S2C). Pathways related to metabolism enriched in the downregulated gene set included "*glycerophospholipid metabolism*," "*inositol phosphate metabolism*," and interestingly "*insulin signaling pathway*" (Fig. S2C).

Insulin supplementation alone tended to increase expression of key PUFA biosynthesis genes except for *Δ5fad*, which did not significantly ($q < 0.05$) change with increasing insulin concentration (Fig. 4D). Increasing insulin concentration from 10 to 100 nM did not significantly change the expression of any of the five genes. Addition of 70 μM ALA had a large effect on the expression of *Δ5fad* and *Δ6fada*, which were expressed most in the absence of insulin, then downregulated upon insulin supplementation. ALA supplementation did not appear to have a large effect on the expression of *elovl5a* and *elovl5b*, which agrees with findings from the ALA gradient experiment.

## Liver slice culture versus primary cell culture

To assess how liver slice culture compares to widely used hepatocyte culture and liver in vivo, we compared RNA sequencing data from hepatocyte culture ($n = 16$ from one fish), liver slice culture ($n = 89$ from four fish), and whole liver ($n = 210$ from 210 fish). Hepatocytes were sampled after 5, 6, and 8 days in culture, so only liver slices incubated at least 5 days were used for comparison. The hepatocytes displayed a cuboidal morphology and we did not find an increase in the expression of the viability marker apoptosis inducing factor 1 (*aif1*, Fig. S3) or a reduction in RNA quality or abundance over time, indicating that primary hepatocytes remained healthy after 8 days in culture. Data on whole liver was obtained from a feeding trial where salmon were fed either a fish oil or plant oil based diet (*Gillard et al., 2018*). All data from each source was pooled to give a range of possible expression patterns from that source. We then calculated relative expression by scaling expression of each gene across all data sources.

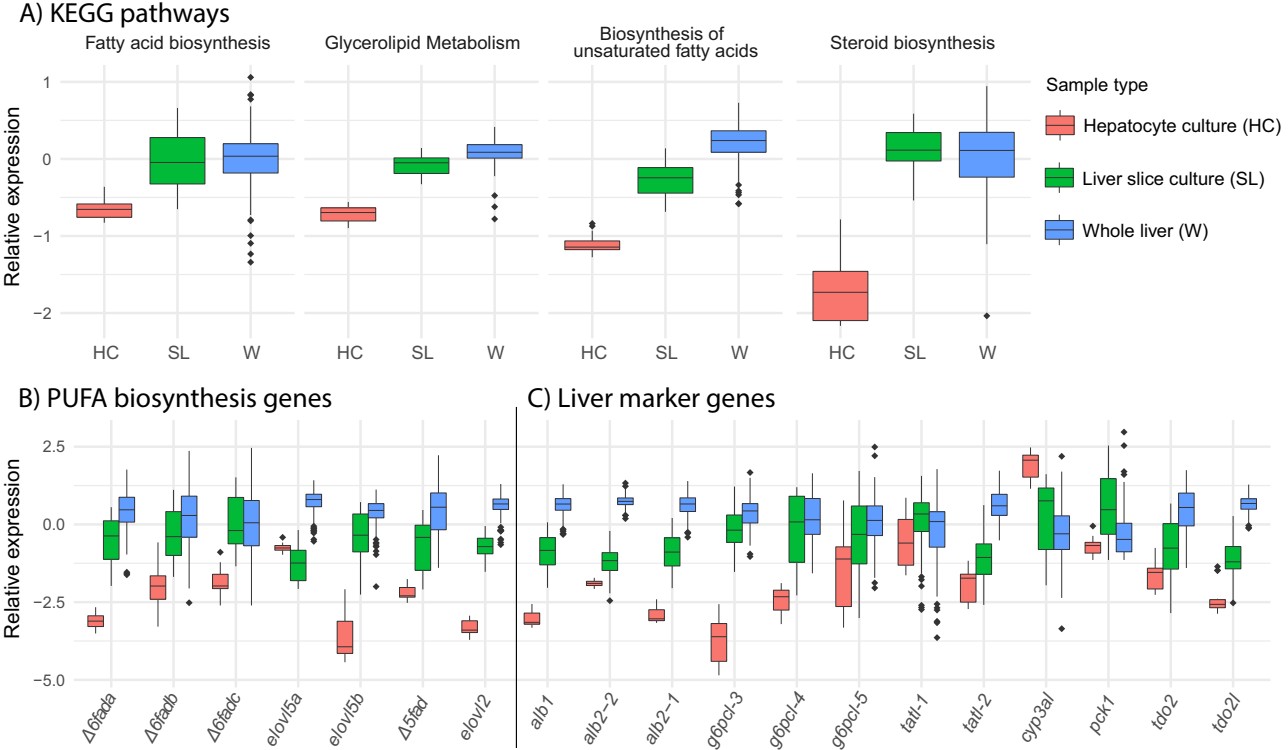

**Figure 5 Expression of select pathways and genes in hepatocyte culture, liver slice culture, and whole liver.** Transcriptomic data from each source was pooled to give a range of possible gene expression levels (hepatocyte culture $n = 16$, liver slice culture $n = 89$, whole liver $n = 210$). (A) Relative expression profiles for five selected lipid metabolism pathways. Values are expressed mean scaled log2 CPM of all genes within a pathway for each sample with a minimum CPM of 1. (B) Relative expression of key genes in the PUFA biosynthesis pathway with a minimum CPM of 10. Abbreviations: *Δ6fad*, delta-6 fatty acid desaturase; *elovl5*, fatty acid elongase 5; *Δ5fad*, delta-5 fatty acid desaturase; *elovl2*, fatty acid elongase 2. (C) Relative expression of select liver marker genes with a minimum CPM of 10. Abbreviations: *alb*, albumin; *g6pcl*, glucose-6-phosphatase-like; *tatl*, tyrosine aminotransferase-like; *cyp3al*, cytochrome P450 3A27-like; *pck1*, phosphoenolpyruvate carboxykinase 1; *tdo2*, tryptophan 2,3-dioxygenase; *tod2l*, tryptophan 2,3-dioxygenase-like.

We find that overall, liver slice culture more closely resembles whole liver than hepatocyte culture. Gene expression in the lipid related KEGG pathways *"fatty acid biosynthesis,"* *"glycerolipid metabolism,"* *"biosynthesis of unsaturated fatty acids,"* and *"steroid biosynthesis"* was much lower in hepatocyte culture relative to both liver slice culture and whole liver (Fig. 5A). This was reflected in the expression of all underlying key PUFA biosynthesis genes except *elovl5a* (Fig. 5B). Expression of genetic marker genes characteristic of functional liver was generally lower in hepatocyte culture, while liver slice culture was closer, but not identical to whole liver expression patterns (Fig. 5C). Specifically, albumin genes *alb1*, *alb2-1*, and *alb2-2* and glucose-6-phosphatase genes *g6pcl-3* and *g6pcl-4* had lowest expression in hepatocyte culture followed by liver slice culture and highest expression in whole liver (Fig. 5C).

## DISCUSSION

### Liver slice metabolism stabilizes after 3 days and remains liver-like through 9 days in culture

We find that time, up to 9 days tested, does not have a large effect on the viability or thickness of the slices. This is encouraging, since morphometric analysis of Atlantic cod

liver slices showed an increase in the proportion of dead cells at 72 h in culture (*Eide et al., 2014*) and studies on rat liver slices have shown that changes in viability and slice thickness over time is highly dependent on the culture media used (*Starokozhko et al., 2015*).

We do, however, observe a time dependent drift in gene expression patterns. Slices most resemble whole liver 24 h after slicing with a correlation coefficient of 0.90 and gradually decrease in similarity over time (Fig. 3). A similar effect has been observed in rat liver slices (*Boess et al., 2003*), however, the rate that slices diverged from whole liver was much lower in our experiments. High correlation (>0.8) to whole liver was maintained through day 4 in time course one and through day 6 in time courses two and three. The observed decrease in co-expression correlation in the pathway "*steroid biosynthesis*" was due to upregulation of nearly all steroid biosynthesis genes 3 days after slicing and is probably due to a deficiency of cholesterol in the media. BCD is known to dose dependently remove cholesterol from cell membranes in culture, however, since samples were taken before BCD was added on day 3 and co-expression correlation decreases on day 3 (before BCD exposure), it is unlikely that BCD is causing this effect. Genes that are highly upregulated 24 h after slicing were mostly enriched in signaling pathways (Fig. 2), likely related to repair and inflammatory response processes known to be triggered by physical liver damage that is unavoidable during the slicing process (*Su et al., 2002*). Since metabolic gene expression stabilizes after 3 days (Fig. 2, clusters 2 and 3), we used a 3-day recovery period for future metabolic studies so that changes in gene expression are more likely to be the result of the treatment rather than time. The gradual downward trend in co-expression correlation over time represents a slow drift in the global gene expression phenotype as opposed to a rapid gene expression change upon hepatocyte culturing. This is a known problem with hepatocyte cultures resulting from a combination of factors, especially the lack of circulating hormones produced elsewhere in the body causing time-dependent de-differentiation of hepatocytes (*Elaut et al., 2006*). Expression of liver marker genes was markedly higher in our liver slice culture than 2D hepatocyte culture (Fig. 5C) representing an improvement in long-term hepatocyte stability.

### Exogenous ALA is taken up and triggers a liver-like response

Alpha-linolenic acid complexed with BCD was efficiently delivered to cells in a dose dependent manner. We observed a proportional increase in ALA and 20:3$n$-3 with increasing ALA concentration. While the ALA increase could be due to residual ALA sticking to the cells from the media, the proportional increase in 20:3$n$-3 with media ALA concentration supports active uptake and elongation of exogenous fatty acids (Fig. 4C). There was no significant change in EPA or DHA after ALA supplementation. This does not mean that EPA and DHA are not being produced, but rather that the amount of ALA in the media is too low to cause a measurable increase in the already abundant pool of EPA and DHA in the cells. Indeed, both ALA and 20:3$n$-3 are low (0.88% and 0.35%, respectively) in control slices, so a small increase in abundance could be detected. Future studies feeding radiolabeled ALA to liver slices are needed to confirm the production of DHA and EPA in PCLS.

Alpha-linolenic acid fed to slices has two fates within the PUFA biosynthesis pathway. The first and most common is the canonical pathway, where ALA is first desaturated by a Δ6 desaturase to 18:4$n$-3, then elongated and desaturated to EPA and DHA via Sprecher's shunt (*Voss et al., 1991*). The second occurs when ALA is first elongated to 20:3$n$-3, presumably by ELOVL5. In this case, a Δ8 desaturase is required to form 20:4$n$-3, which can then continue to EPA and DHA via the canonical pathway. This does, however, not happen efficiently in Atlantic salmon because of the low Δ8 desaturase activity of Δ6FADb (*Monroig, Li & Tocher, 2011*) in combination with low expression in liver (0.5–3.2 CPM). Rather, 20:3$n$-3 accumulates in the cells or is catabolized for energy (*Tocher, 2003*), which is consistent with observations in feeding trials where fish fed vegetable oil based diets high in ALA contained higher tissue levels of 20:3$n$-3 (*Tocher et al., 2001*; *Bell et al., 2010*). This can explain why we measure increased levels of 20:3$n$-3, but not other PUFA intermediates.

We also observe that 18:0 increases between whole liver and liver slices. This corresponds with increased expression of both fatty acid synthase genes, *fasa* and *fasb*, which doubled (log2FC 1.08 and 1.02, respectively) in expression 1 day after slicing and the pathways "*fatty acid biosynthesis*" and "*glycerolipid metabolism*" were enriched in clusters that spike 1 day after slicing (clusters two and seven, Fig. 2C). Conversely, high levels of saturated fatty acids are known to be toxic to cells, so to avoid this cells rapidly desaturate 18:0 to 18:1$n$-9, but we observe a decrease in this fatty acid in slices. This suggests that de novo lipogenesis is not responsible for the increase in 18:0. Upregulation of *fas* could be stress related since *fas* is known to play a role in remodeling O-GlcNAcetylation patterns during oxidative stress and injury (*Groves et al., 2017*). It is also unlikely that the 18:0 originates from FBS added to the media, since previous studies have found that FBS also contains high levels of 18:1$n$-9 and we find that this decreases in liver slices (*Stoll & Spector, 1984*). In addition, we found that as media ALA concentration increased, total fat in liver slices decreased (Table S2). This corresponds with BCD concentrations of 700 μM in control slices (zero μM ALA) and between 200 μM (20 μM ALA) up to 1,400 μM (140M ALA). Since BCD concentrations are the same in control slices and slices fed 70 μM ALA, but total fat is lower in 70 μM ALA slices, it appears that this effect is related to the concentration of ALA rather than the concentration of BCD. It is possible that feeding high levels of ALA to liver slices increases oxidative stress and induces lipolysis, thus decreasing total fat in a dose dependent manner. Alternatively, ALA could stimulate lipoprotein production in liver slices, similarly reducing total fat in slices. Future experiments could measure lipoprotein formation and markers for oxidative stress in liver slices fed high levels of ALA to explain this effect.

Overall, ALA concentration was positively correlated to lipid metabolism related gene expression, especially in PUFA biosynthesis with expression of all key pathway genes increasing with ALA. The same effect is known to occur in Atlantic salmon livers where fish fed vegetable oil-based diets high in ALA have higher PUFA biosynthesis gene expression relative to salmon fed fish oil-based diets low in ALA and high in EPA/DHA (*Gillard et al., 2018*). Additionally, this has been observed in vitro using Atlantic salmon primary hepatocytes (*Kjær et al., 2016*) and in vivo on rat liver (*Tu et al., 2010*). At very low

concentration (20 µM), expression of *Δ5fad*, *Δ6fada*, and *elovl5b* actually decreased relative to control slices with no ALA supplementation. In this experiment samples were taken 2 days after exposure to ALA, so it is possible that in 48 h all of the ALA in the media was depleted, presumably taken up by the cells and anabolized to longer chain fatty acid products that have an inhibitory effect on expression. Additionally, the "*PPAR signaling pathway*," which includes PPARs and target genes, was significantly enriched in cluster three which increases with increasing ALA concentration (Fig. S1). PPARs are well known transcriptional factors that bind fatty acids and in turn activate genes involved in a wide range of cellular functions, most notably lipid metabolism (*Poulsen, Siersbæk & Mandrup, 2012*). Taken together our results demonstrate the ability of our PCLS model to accurately mimic expected shifts in lipid metabolism genes, highlighting its quality as an in vitro system.

## Insulin triggers an anabolic response

Lipid metabolism, like other metabolic processes, is highly influenced by the feed status of the fish with insulin production triggered by feeding (*Navarro et al., 2002*). In order to ensure that the liver slices behaved similarly to liver in fed fish, we assessed the inclusion of insulin in the media. A main function of insulin is to shift the metabolic state from catabolic to anabolic, since it would be counterproductive for cells to actively produce energy by breaking down organic macromolecules while at the same time storing energy by building them up (*Dimitriadis et al., 2011*). In line with this we observe a binary response with several thousand genes either upregulated or downregulated in the presence of insulin, regardless of concentration (Fig. S2). Major anabolic pathways including "*biosynthesis of unsaturated fatty acids*" and "*fatty acid biosynthesis*" are upregulated in the presence of insulin in agreement with an anabolic response. Physiological range for circulating insulin is 0.2–5 nM (*Caruso & Sheridan, 2011*), so it is plausible that raising insulin concentrations to 100 nM has little effect because all of the insulin receptors are likely bound at 10 nM.

## Insulin and ALA interact to regulate PUFA biosynthesis gene expression

Insulin and ALA displayed complex interaction effects on expression of genes related to PUFA biosynthesis in liver slices. Unaffected by ALA concentration, *elovl5a* and *elovl5b* were highly upregulated in the presence of insulin. On the other hand, genes that were upregulated in response to ALA tended to be upregulated in the presence of insulin alone, but then downregulated in the presence of insulin when combined with ALA (Fig. 4D). An important regulator of lipid metabolism in liver, sterol regulatory element binding protein 1 (*srebp-1*), is known to be upregulated by insulin through the PI3K/Akt/mTOR signaling pathway (*Matsuzaka & Shimano, 2013*), and indeed *srebp-1* is upregulated in response to insulin in our experiments. Both *elovl5a* and *elovl5b* contain sterol regulatory elements in their promoter regions (*Carmona-Antoñanzas et al., 2013*), and along with *Δ6fada* have been shown to increase in expression when co-transfected with *srebp-1* (*Carmona-Antonanzas et al., 2014*). On the other hand, activation of PPARα by ALA

could work in opposition to insulin-mediated effects by stimulating beta-oxidation and ketogenesis. There is evidence in rats that Δ5 desaturase (D5D) and Δ6 desaturase (D6D) are under dual regulation by both SREBP-1 and PPARα (*Matsuzaka et al., 2002*), and given that regulation of lipid metabolism is highly conserved across species (*Carmona-Antonanzas et al., 2014*) it is likely a similar effect is present in salmon. The contrasting effect of insulin and ALA supplementation highlights the complex interplay between signaling networks balancing hormonal and nutritional input to optimize regulation of PUFA metabolism in Atlantic salmon.

## Liver slices are more suitable for long-term culture than primary hepatocytes

We find that liver slices maintained liver-like gene expression patterns for longer than primary heptatocytes, which are generally known to undergo time-dependent de-differentiation (*Elaut et al., 2006*). We attribute this to the maintenance of the complex three-dimensional organization of whole liver with all interacting cell types. While hepatocytes are generally responsible for the metabolic activities associated with liver, regulation of these functions is controlled in concert with nonparenchymal cells through complex endocrine and autocrine signaling networks (*Kmiec, 2001*). Eicosanoid signaling is a key component of these networks, which represents a layer of information that is completely lost in 2D hepatocyte cultures since eicosanoids are only produced in nonparenchymal cells (*Johnston & Kroening, 1996*). In mammals, glucose metabolism has been demonstrated to be influenced by nonparenchymal produced eicosanoids (*Cherrington, 1999*) and there is evidence that regulation of lipogenesis and PUFA metabolism is influenced by eicosanoid-mediated effects (*Jump et al., 1999*). In addition to eicosanoid production, interactions between hepatocytes and nonparenchymal cells are known to play a role in cell proliferation and differentiation (*Kmiec, 2001*) which could explain the higher liver slice culture expression of liver marker genes. This, along with many other factors likely contribute to the observed long-term differences between liver slice culture and 2D hepatocyte culture. Many of the metabolic processes in the liver are also regulated by circulating hormones produced in other parts of the body, so while liver slice culture is not identical to whole liver, we find that liver slice culture is better suited to long term metabolic studies than primary hepatocyte culture.

## CONCLUSION

Taken together, our results demonstrate the utility and effectiveness of precision cut liver slices as a tool for studying lipid metabolism in Atlantic salmon. We found that when studying metabolism in liver slices, it is best to allow the slices to recover for 3 days before adding fatty acids, since gene expression in pathways relating to metabolism remains stable after 3 days in culture. Liver slices were highly responsive to both exogenous fatty acids and insulin in line with current understanding of lipid metabolism of Atlantic salmon. Supplementation with ALA induced expression of lipid metabolism genes and pathways while supplementation with insulin shifted gene expression to an anabolic state as expected. We also observed a different, sometimes opposing, regulatory effect of insulin

and ALA on expression of genes involved in PUFA biosynthesis. Liver slices mimic the complex three-dimensional structure of the liver and produce results that are more relatable to liver in vivo than 2D hepatocyte culture. For this reason, liver slices are an attractive alternative to 2D hepatocyte culture for interrogating metabolic pathways.

### Funding
This work was funded by projects DigiSal NFR 248792 and GenoSysFat NFR 244164. The funders had no role in study design, data collection and analysis, decision to publish, or preparation of the manuscript.

### Grant Disclosures
The following grant information was disclosed by the authors:
DigiSal NFR 248792 and GenoSysFat NFR 244164.

### Competing Interests
Jacob S. Torgersen is employed by AquaGen AS.

### Author Contributions
- Thomas N. Harvey conceived and designed the experiments, performed the experiments, analyzed the data, contributed reagents/materials/analysis tools, prepared figures and/or tables, authored or reviewed drafts of the paper, approved the final draft.
- Simen R. Sandve conceived and designed the experiments, contributed reagents/materials/ analysis tools, authored or reviewed drafts of the paper, approved the final draft.
- Yang Jin analyzed the data, authored or reviewed drafts of the paper, approved the final draft.
- Jon Olav Vik conceived and designed the experiments, contributed reagents/materials/ analysis tools, authored or reviewed drafts of the paper, approved the final draft.
- Jacob S. Torgersen conceived and designed the experiments, performed the experiments, contributed reagents/materials/analysis tools, authored or reviewed drafts of the paper, approved the final draft.

### Animal Ethics
The following information was supplied relating to ethical approvals (i.e., approving body and any reference numbers):

Atlantic salmon used in this study were treated according to the Norwegian Animal Research Authority (NARA) in accordance with the Norwegian Animal Welfare Act of 19th of June 2009.

### Data Availability
Data is available at ArrayExpress under accession numbers: E-MTAB-7368, E-MTAB-7367, E-MTAB-7366 and E-MTAB-7364.

## Supplemental Information

Supplemental information for this article can be found online at http://dx.doi.org/10.7717/peerj.7732#supplemental-information.

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
