# Peer review of "Liver slice culture as a model for lipid metabolism in fish"

_PeerJ, doi:10.7717/peerj.7732_

## Round 0.1 · original submission · Major Revisions

· Academic Editor

Major Revisions

This manuscript provides interesting new information and data about the use of a liver slice culture, as compared with hepatocyte culture, to study and evaluate lipid metabolism in salmon. The reviewer comments were generally positive. However, the reviewers have made many comments, both major and minor, that need to be addressed to clarify some of the methods and justify some choices made in the experimental design. These will provide greater clarity to readers. Please address all of the points raised by the reviewers and clearly indicate where revisions were made in the text.

Reviewer 1 ·

Basic reporting

1. Basic Reporting
The quality of writing is quite good, only a few minor editorial suggestions (e.g. L17 “…interacting cell types essential for liver function.”)

L26: not sure what “…exceedingly more similar to whole liver..” means? This is not a typical comparator comment. Suggest an alternate wording to indicate that PCLS profiles, as measure in this study, are statistically, more similar to whole liver than that of hepatocytes. I think this is what is meant here.

L100 & 106: .”..(empty BCD)..” suggest change to “(BCD only)” for clarity
Figure legends do not indicate n values (see comments under Experimental design section).

L107. Assume this refers to ALA concentration gradient? “..70, 100 and 140 on day three ..” units missing.

Figure Legends in general, require some additional detail for clarity.
Figure 1: no indication what the individual symbols represent in 1B…a single liver slice?? In panel C, no labels to indicate important features. L191-192, no mention of features to support this statement.
Figure 4: it would be helpful to indicate n values on which the means are based..yes, the individual data are shown as separate circles, but difficult to assess, especially if some overlap. Also, spell out abbreviations.

L253 Figure 4C is not mentioned prior to Fig 4D..rather the former is not mentioned until L279. Suggest changing figure order or mention order to correct this.
L268: “…(FDR <0.01; Fig 5A)” This is not the correct reference…I think it should be Supplemental Fig 2A.
L327: there is no Fig 6C.

Experimental design

2. Experimental Design
The research in question is well defined and will assist in the fields of piscine toxicology and aquaculture moving forward, as it relates to enhanced throughput for feed and toxicity assessment. The rationale is well described and convincing.

The experiments performed are appropriate, as is the analysis. The text describing the methods is somewhat confusing and lacks some crucial detail, as outlined below.

The Methods section is rather confusing as written and some details are missing.
L 77..descrioption of culture method lacks information about ambient gas content…were the cultures in air? Or were they incubated in conditions typical for mammalian culture (5%CO2:95%O2)? This is important, as it may impact the acid-base status of the culture medium. The latter was not mentioned...what was the pH of the media? Representative of Atlantic salmon ECF at 15°C?

It was also not clear why salmon of different sizes (50g or 500g) were used, and why freshwater and seawater fish were used. It wasn’t clear if these differences may have confounded the data, especially those reported in Fig 5.

Determining the n value for each experiment was most difficult…it was not clear whether the liver was taken from a single fish, and multiple slices were obtained and thus, n refers to slice number, not number of fish. (if so, then, arguably, then data represent technical, not biological replicates.). The figure legends do not report n values. This needs to be addressed.

L113-114: How were livers injected? Via a blood vessel?
L114: type of collagenase? There are many different kinds.
L117: pellet likely resuspended, rather than dissolved.
L117: Why different source of insulin (Merk vs. Sigma). Human insulin?
L125-126: how were sections of liver slices chosen for imaging and assessment?
L128: How many liver slices were examined?
L172: what minor changes?

Validity of the findings

The authors are to be congratulated on collecting and analyzing a comprehensive data set. The conclusions drawn are well supported by the data. It is convincingly argued that PCLS is far superior to 2D hepatocyte culture for studying hepatic function. Interestingly, the insulin and ALA interactions were not as predicted...which opens an new window for investigation.

One disconnect (at least for me) are in the steroid biosynthesis data. In Fig 5A, the PCLS are shown to have similar expression as whole liver, and that appears to contradict what is shown in Fig 3, where steroid biosynthesis pathway expression appears to be significantly reduced compared to intact liver.

L388: “…all of the insulin receptors are bound at 10 nM.” Suggest softening here…to “are likely bound”.

Reviewer 2 ·

Basic reporting

no comment

Experimental design

Methods:
• It is stated (line 68) that the study was treated according to the Norwegian Animal Research Authority in accordance with the animal welfare act of June 2009.

The experiment should be in compliance with the “newer” national (FOR-2015-06-18-761) regulation for use of experimental animals (which is in agreement with EU regulation (Directive 2010/63/EU), it is also recommended that the authors include a reference to their FOTS approval of the experiment.

• The authors compare gene expression between whole liver, liver slice and hepatocyte culture by using pooled data from each source to give an overall expression phenotype.

Livers are sampled from fish in different life stages freshwater and saltwater and fed different diets, these are all factors that will have major impact on the expression of metabolic gene. The samples to be compared should ideally come from fish fed the same diets, from same life stages in order to be able to make a sound comparison. The authors need to give more information to justify that their comparisons are sound.

• Lipid analyses:
The authors state that they used 13:0 as an internal standard for fatty acid composition analyses.
It is not very common to use this short chain fatty acid, when there is specific emphasis on determining the level of the longer chain fatty acids. Why have the authors used 13:0 and not the more relevant longer chain 23:0? Further in their GC analyses, no effect on the relative proportions of EPA and DHA with increasing ALA inclusion in the media was found but instead a tendency to reduced EPA + DHA. This reduction is most likely due to a dilution effect when ALA increase. Although 13:0 is maybe not the best std to calculate the quantitative levels of EPA and DHA, the authors are recommended to give these quantitative values for the fatty acids in addition to the relative (to see if the actual quantity of fatty acids change when ALA increase)..

• Hepatocyte culture.
• It is stated that the cells were grown at 200K density, the authors need to give the density per area. Further the authors are recommended to include the fatty acid composition of the livers used in the study for isolation of hepatocytes in addition to the life stage, since these factors will influence the gene expression in the cells.
.
• Isolated primary hepatocytes in culture are normally used for metabolic studies within the first 4 days after isolation and cultivation, thereafter the cells will start to lose their “hepatocyte” characteristics/nature/function and become more and more fibroblast like cells. The authors need to explain why they have chosen to compare the expression of metabolic genes between day 5 to day 9, at these time points the cells are not normally suitable for such comparisons. While the gene expression in liver slice culture stabilize after 4 days and the authors consider them suitable for metabolic studies between day 4 and 9, this is normally not the case for cultivated hepatocytes.

Validity of the findings

Viability measurements

• It is only measured dead and alive cells as test of viability, ideally it should be included a measurement of the viability of live cells, for instance by measuring mitochondrial function MTT, or LDH leakage, since live cells do not necessarily need to be “healthy”. It is stated that the liver slices were highly viable in all experiments with a mean viability of 90%. When looking at supplementary figure S1., there is a major increase in expression of genes related to autophagy, mitophagy, proteasome and lysosome in cells supplemented with insulin and the two highest doses of ALA. Autophagy can be seen as an response to cellular stress, which can promote cell death or in extreme cases, the breakdown of cellular components promotes cellular survival by maintaining cellular energy levels showing the need to include more thorough viability evaluations of the cells.
The combination of insulin and ALA led to downregulation of many metabolic genes that normally are upregulated by insulin and ALA, the authors need to evaluate if this is caused by a potential toxic effect by the high doses of ALA and insulin. Stress in cells may lead to downregulation of a huge number of genes.

• Viability measurements of hepatocytes in culture also needs to be included, also a test showing the viability from day 1 after isolation to day 9 at the end of the experiment, It is further recommended to include a microscopy image of the hepatocytes in culture (in supplement), to visualize that the cells actually have formed bridges (mimicking) liver structure, which healthy salmon liver cells do in culture

De novo lipogenesis
• Line 259, it is stated that there was a large difference (doubling) in proportion of 18:0 between fresh liver and liver slices after 4 days of incubation. It is further discussed in line 357 that these fatty acids are synthesized de novo. However, it is believed that the capacity to produce fatty acids de novo in salmon is relatively limited and it should be more thoroughly discussed why the authors believe this?

Discussion of liver slice culture versus primary cell culture

• It is concluded that liver slice culture outperforms 2D hepatocyte culture in terms of gene expression and similarity to whole liver. However, the authors are recommended to modify their conclusion since the experimental conditions do not justify a comparison of the two model systems. Further there is a huge amount of studies on 2D hepatocyte culture that show their suitability to study metabolism of fatty acids and conversion to all metabolic products in the omega-3 pathway, whereas in the liver slice culture in this study, only 20:3n-3 could be found.

Additional comments

The manuscript “Liver slice culture as a model for lipid metabolism in fish (#34550)” present some new and interesting data on how liver slice culture could be used as a model for studies on lipid metabolism in fish. However, some of the conclusions drawn could not be thoroughly justified.

---

## Round 0.2 · Major Revisions

· Academic Editor

Major Revisions

As you can see, reviewer #2 still thinks that some of the critiques are unanswered. Please carefully address these remaining critiques and discuss possible shortcomings of your data.

Reviewer 2 ·

Basic reporting

No comment

Experimental design

No comment

Validity of the findings

Comments to steroid biosynthesis results:
Fig 3, shows co-expression correlation between mean whole liver expression and gene expression in liver slice from different days. The steroid biosynthesis pathway expression appears to be significantly reduced in liver slice (from day 3-9) compared to intact liver. In primary hepatocytes, many genes in the steroid biosynthesis pathway was lower than both that in both liver slice and whole liver.
Based on these results, the authors conclude that the liver slices were closer to whole liver expression relative to primary hepatocytes.
However, the authors need to consider the potential influence of the use of methyl-β-cyclodextrin (MβCD) in their model. MβCD is an effective agent for the removal of plasma membrane and mitochondrial membrane cholesterol. Several studies have shown that MβCD dose-dependently increase the cholesterol level in the culture medium and the intracellular level of free cholesterol, possibly through removal of cholesterol from the plasma membrane (even when a non-toxic concentration of MβCD is used). In this study it was used 0,09% MβCD in the incubation over 9 days. Even though less than 0,1% MβCD was used, which is considered safe dose for cells in short term cultures, detrimental effects may still occur during longer term incubations (as 9 days in this study). It has also been proposed that the variability of the toxic effects of MβCD observed in different cell system studied is correlated with the concentration of cholesterol in the membrane. For instance, 0.2% MβCD toxicity in fibroblasts cell lines (Pfitzner et al., 2000) exhibit lactate dehydrogenase activity in media culture comparable to control. However, 1% concentrations of MβCD triggered a significant induction of cell death and apoptosis in human keratinocytes cell lines (Schonfelder et al., 2006). Cancer cells are also shown to be more sensitive to cholesterol depleting agents because their membranes are more enriched with cholesterol. Salmon cells are known to be particularly rich in cholesterol, which may suggest a high sensitivity in this species for MβCD. Further, MβCD is a well‐known lipid microdomain disrupting agent and cholesterol chelator and it is shown to disrupt mitochondrial raft‐like microdomains by cholesterol efflux and impair mitochondrial bioenergetics.
Thus, the authors need to discuss how sensitive the salmon liver cells are towards MβCD in longer term exposure? If cholesterol is released from plasma membranes and mitochondrial membranes, it may explain the downregulation of genes in the sterol biosynthesis.
.
Previous reviewer comment to De novo lipogenesis results
Line 259, it is stated that there was a large difference (doubling) in proportion of 18:0 between fresh liver and liver slices after 4 days of incubation. It is further discussed in line 357 that these fatty acids are synthesized de novo. However, it is believed that the capacity to produce fatty acids de novo in salmon is relatively limited and it should be more thoroughly discussed why the authors believe this?
Authors comments
While it is true that de novo lipogenesis (DNL) is thought to be active but limited in salmon, our data suggests a large increase in DNL one day after slicing. Expression of fasa and fasb more than doubled (log2FC 1.08 and 1.02 respectively) between whole liver (D0) and day 1. Additionally, L15 is high in D+ galactose (5mM) and sodium pyruvate (5mM) providing ample supply of acetate precursor. It is also possible that the 18:0 originates from FBS added to the media, however previous studies have found that FBS also contains high levels of 18:1n9 and we find that this decreases in liver slices.
We have added this to the discussion on lines 353-360.

New comment reviewer:
The major products from de novo lipogenesis are known to be 16:0 and 18:0. Since an increase in these saturated fatty acids are known to be very toxic for cells, the hepatocytes have a huge capacity to desaturate these fatty acids further to 16:1n-7 and 18:1n-9, since MUFA prevent toxic effects. However, when looking at the quantitative levels mg/100 gram of these FAs, the levels of both 16:0, 16:1n-7 and 18:1n-9 are reduced, which do not point in direction of increased de novo lipogenesis, since it is only the level of the saturated fatty acid 18:0 that increases. Further data is therefore needed if in order to conclude that increased de novo lipogenesis is the cause of the increased 18:0 level.
The authors point to that fasa and fasb more than doubled (log2FC 1.08 and 1.02 respectively) between whole liver (D0) and day 1, indicating increased lipid synthesis. However then one would expect increase in total liver slice lipid content (which is the opposite of what is shown in Table S2). However, FAS gene expression may also increase during stress, for instance FAS induction in stressed myocardium represents a compensatory response to protect cardiomyocytes from pathological calcium flux. Also, data suggest that FAS fine-tunes the cell's response to stress and injury by remodeling cellular O-GlcNAcylation.
Table S2 shows that the supplementation of ALA leads to a major reduction in the total fatty acid level in liver slice cells. The level is reduced from 3.20-gram fatty acid per 100-gram biomass (in the 0 ALA group) to 1.83 fatty acids per 100-gram biomass in the 140 ALA group, even though this group has taken up ALA FAs. Why has there been an approximately 50% loss of cellular fatty acids when increasing dose of ALA is given to the cells? Can this loss of fatty acids may also indicate increased lipolysis due to toxic effects or increased dose of MβCD resulting in leaky membranes?
Further, disruption of microdomains by cholesterol extraction with methyl-β-cyclodextrin may inhibit tyrosine phosphorylation and insulin-induced glucose transport activation. It is shown that the cholesterol-lowering agent methyl-β-cyclodextrin promotes glucose uptake and reduces insulin resistance in obese mice. The authors need to discuss if MβCD in the culture media may be one factor influencing the confounding effects of ALA and insulin in their study.

Additional comments

The Authors state that liver slices were highly viable in all Experiments. However, there are some concerns concerning the validity of their findings that the Authors needs to address/discuss

---

## Round 0.3 · accepted · Accept

· Academic Editor

Accept

Thank you for addressing the remaining critiques of reviewer #2.